

# 1 SODA4: a mesoscale ocean/sea ice reanalysis 1980-2024

Gennady A. Chepurin[1], James A. Carton[1], Luyu Sun[1], and Stephen G. Penny[2]
[1]Department of Atmospheric and Oceanic Science, University of Maryland, College Park, Maryland, 20742 USA
[2]Sofar Ocean Technologies, 28 Pier Annex, San Francisco, CA 94105 USA
*Correspondence to*: James A. Carton (carton@umd.edu)
**Abstract.** This paper describes the new Simple Ocean Data Assimilation version 4 (SODA4) global eddy-resolving
ocean/sea ice reanalysis that spans the 45-year period 1980-2024. The reanalysis is constructed using GFDL MOM5/SIS1
numerics and ECMWF ERA5 forcings with surface and subsurface temperature and salinity observations as constraints
within an optimal interpolation data assimilation algorithm. The method of construction and resulting output files are
described. Comparison of the SODA4 temperature and salinity fields to observations and to the UK Met Office EN4
temperature and salinity analyses in the upper ocean shows SODA4 has marginal bias and exhibits more regional variability,
with less of an imprint of the sparse and inhomogeneous distribution of observations. Comparison of transports across major
ocean sections and passages are generally consistent with independent moored observations.

## 16 1 Introduction

We describe the Simple Ocean Data Assimilation version 4 (SODA4) eddy-resolving reanalysis of the global physical state
of the ocean/sea ice system during the 45-year period 1980-2024. Section 1.1 provides a brief history of the SODA project.
Sections 2 and 3 describe the model and forcing, data assimilation, constraining data sets, and output files. Section 4
examines mean and variable temperature and salinity fields in comparison to the no-numerical forecast model EN4.2.2
monthly subsurface temperature and salinity analysis of Good et al (2013). Mean volume transports through some major
passages are examined in comparison to moored estimates.

## 24 1.1 Background

The SODA reanalysis project began in the 1980s as an effort to provide a uniformly gridded estimate of the ocean state
during the 1983-4 US/French Seasonal Francais Ocean et Climat dans l'Atlantique/Response of the Equatorial Atlantic
(FOCAL/SEQUAL) experiment in the tropical Atlantic Ocean (*Carton and Hackert, 1989; Carton and Hackert, 1990;*



*Raghunath and Carton, 1990*). From the beginning, SODA was designed to leverage community standard ocean models and
simple data assimilation algorithms previously developed and tested for climate and weather forecasting applications. SODA
gradually expanded to a quasi-global domain following the release of the NCAR/NCEP global atmospheric reanalysis
(Kalnay et al., 1996), then using the 5.3 million hydrographic observations contained in the World Ocean Database 1998
(*Levitus et al., 1998*) as constraints.  This effort culminated in two papers, *Carton et al. (2000a,b)* describing the first global
version of the SODA reanalysis.  Examination of SODA identified several problems (*Chepurin and Carton, 1999*).  Model
resolution was too coarse to resolve key processes. The constraining ocean observations were too sparse and
inhomogeneous, and finally surface forcing errors introduced spurious long-term trends in properties such as heat storage.
During the 2000s SODA2 was reformulated using Parallel Ocean Program numerics with an improved resolution global
0.4°x0.25°x40L displaced pole grid. Daily ECMWF ERA40 forcing (*Uppala et al., 2005*) replaced the monthly
NCAR/NCEP global atmospheric reanalysis and a bias correction scheme was introduced (*Chepurin et al., 2005; Carton*
*and Giese 2008*).  In 2018 the model was upgraded again, this time to the Modular Ocean Model (MOM5/SIS1)
0.25°x0.25°x50L, similar to the ocean/sea ice components of the GFDL CM2.1 coupled model (*Griffies, 2012; Delworth et*
*al., 2012*). The hydrographic observation set was replaced with the 2018 release of the World Ocean Database (*Boyer et al.,*
*2018*).  A new bias-correction scheme was implemented, and an ensemble of reanalyses were produced using a variety of
surface forcings, including ERA5 (*Hershbach et al., 2020*).  The result was the eddy-permitting SODA3 reanalysis of *Carton*
*et al (2018a,b)*.  Despite these improvements some problems remained.  For example, representation of boundary currents
such as the Gulf Stream were too wide and slow, and lacking down gradient eddy momentum transfer.  Shallow warm/fresh
layers associated with summer heating, river discharge, and ice melt were too deep.  SODA4 has been developed to address
these limitations by increasing model resolution to 0.1°x0.1°x75L, upgrading the observation sets to include the 18.6 million
profiles contained in the World Ocean Database 2023, and adding improved estimates of continental discharge (*Mishonov et*
*al., 2024*).  Comparison of SODA3 and SODA4 surface velocity and eddy kinetic energy **Fig. 1** illustrates the impacts of the
enhanced resolution.





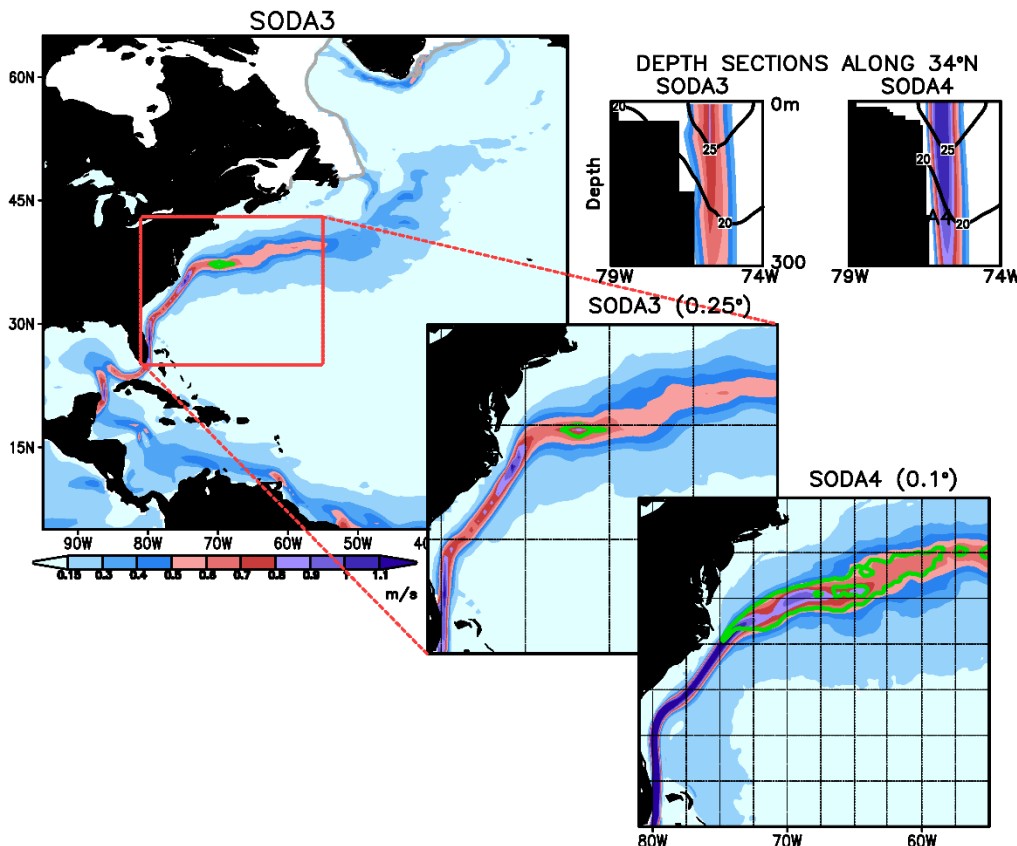

**Figure 1** Comparison of time-mean surface currents in the North Atlantic 2015-2023. (upper left) eddy-permitting SODA3 (0.25º×0.25º×50L), (lower right) eddy-resolving SODAA4 (0.1°×0.1°×75L). (colors) speed, (green contours) eddy kinetic energy. Every 25th gridline is shown.


## 2 Model, data assimilation, and observations

### 2.1 Model

The global ocean/sea ice model, which is similar to the ocean and sea ice components of the GFDL CM2.6 coupled model (*Winton* et al., 2014; *Griffies et al.,* 2015), uses MOM5.1/SIS1 numerics (*Winton,* 2000; *Griffies*, 2012) with 3600x2700 eddy resolving quasi-isotropic horizontal grid cells on an Arakawa B-grid with cell sizes varying from 11 km at the equator to 5.5 km at ±60°. Further northward, the Arctic cap splits into two geographically displaced poles located on the Eurasian and North American continents. (**Fig. 2**). For much of the global domain this resolution satisfies the requirements of



*Hallberg* (2013) to be considered 'eddy resolving' but reduces to 'eddy permitting' in the Arctic due to the decrease in the
eddy length-scale.

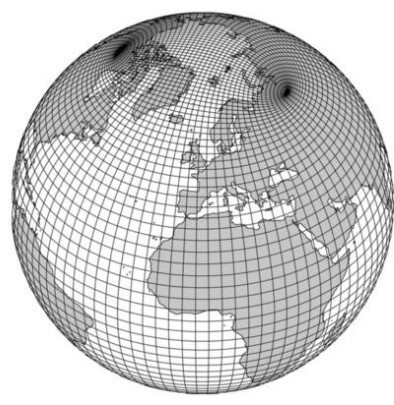

**Figure 2** Tripolar (3600x2700) horizontal grid .  Every 40$^{th}$ gridpoint shown.  Note that grid cells are approximately isotropic. credit: Mats Bentsen, GFDL.


Bottom topography is interpolated from the 30 arcsecond GEBCO 2014 topography (https://www.gebco.net/) with
modifications to eliminate orphan points (**Fig. 3**).  In addition, we eliminate bays spanned by a single grid point and impose
a minimum water depth of 10m, all to prevent numerical problems. A comparison of this topography to the latest GEBCO
2024 topography (Supplementary Materials **Fig. S1)** shows differences that have small geographic scales. The model grid
has 75 z* vertical levels that expand from a finer ~1.1 m resolution near-surface to a coarser 200m resolution in the deep
ocean (Supplementary Materials **Table S1**).  Partial bottom cells are included to improve the fidelity of topographic effects.



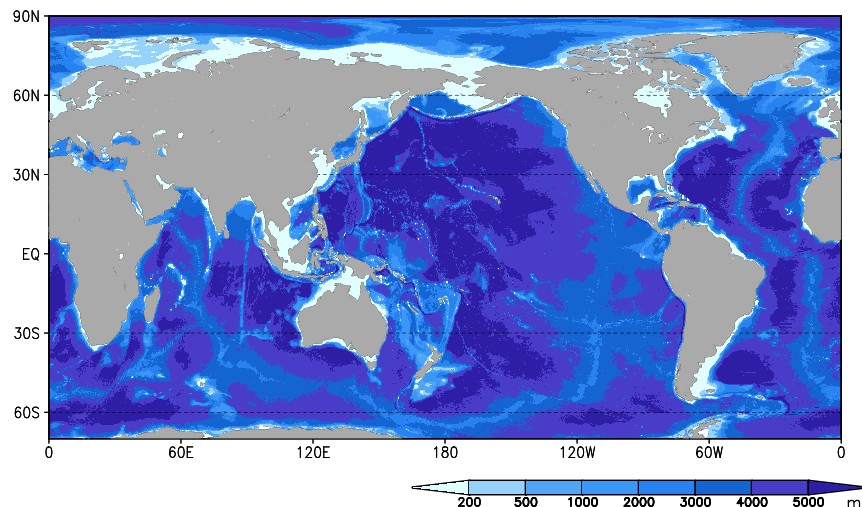

**Figure 3** Basin topography based on the 30 arcsecond GEBCO 2014 topography. Comparison to GEBCO 2024 is shown in Supplementary Materials **Fig. S1**.


The velocity advection scheme was changed during the course of the integration.  We switched among a third order Adams–
Bashforth scheme with a third order upwind biased scheme (*Hundsdorfer and Trompert*, 1994) and monotonicity-preserving
schemes of Daru and Tenaud (2003) to address numerical stability issues.  A predictor-corrector time-filter was applied to sea
level (*Griffies* 2004) to improve numerical stability. Vertical turbulent viscosity varies from $1\times10^{-4}$ to $2.5\times10^{-3}$ $m^2s^{-1}$, while
vertical diffusivity varies from $1\times10^{-5}$ to $5\times10^{-3}$ $m^2s^{-1}$. Additional vertical mixing is added to simulate the impact of tidal
mixing following *Lee et al.* (2006).

Monthly continental discharge is based on our own compilation for this project but is developed from the station gauge
estimates of *Dai and Trenberth (2020)* for a collection of 925 major rivers. *Dai and Trenberth* construct their estimates
primarily from in situ gauge data contained in the Global Runoff Data Centre archive.  Our changes to their estimates
include updating discharge for major rivers in recent years, adding missing discharge from Greenland (*Bamber et al.,* 2018)
and replacing discharge in the Arctic with estimates contained in the Arctic Great Rivers Observatory (*Shiklomanov et al,*
*2020*), corrected for gauge distance upstream of the river delta .  When discharge data is missing we fill the missing values
with climatological monthly discharge computed based on the full record of available observations.  Finally, we adjust time
mean discharge rates basin-by-basin to be in balance with the ERA5 basin evaporation rates in order to bring net freshwater
flux into alignment with basin-average salinity trends.  More details regarding how this was carried out are provided in
Supplementary Materials **Text S1**.  The basin-by-basin discharge rate time series (**Fig. 4)** are dominated by seasonal
variations with the largest discharge as well as the largest year-to-year variations occurring in the Atlantic where a third of
the mean $6.1\times10^5$ $m^3/s$ discharge into the Atlantic basin comes from the Amazon River system (basin definitions are shown




in Supplementary Materials **Fig. S2)**.  More information about discharge for specific rivers is provided in  Supplementary
Materials **Figs. S3-S6**.

Sea ice is modeled using the Sea Ice Simulator (SIS1) of *Winton (2000)* which has horizontal resolution matching the ocean
model.  At each grid point SIS1 specifies five snow and ice categories. Snow albedo is fixed to be 0.85 which lies in the
middle of observational estimates, while ice albedo is set to a high value of 0.8, a value which was chosen to reduce the rate
of summer sea ice melt.  Initial conditions for the ocean and sea ice on 01 January 1980 were interpolated from the previous
SODA3.15.2 eddy-permitting reanalysis.

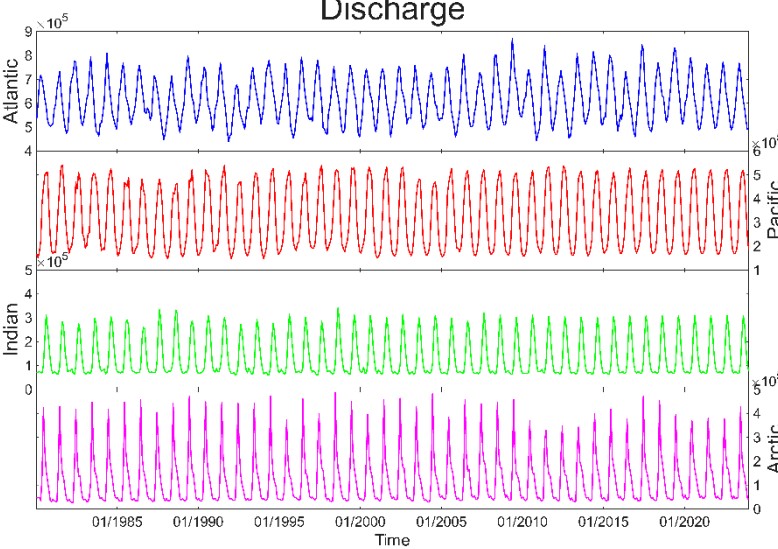

**Figure 4** Continental discharge by basin: (Blue) Atlantic, (red) Pacific, (green) Indian, and (purple) Arctic (note the change in scales). Basin definitions are provided in Supplementary Materials **Fig. S2**.  Time mean values for each basin are: $6.1\times10^5$ m$^3$/s, $3.2\times10^5$ m$^3$/s, $1.4\times10^5$ m$^3$/s, $1.2\times10^5$ m$^3$/s. Global time average discharge is similar to the $1.3\times10^6$ m$^3$/s estimate of *Schmitt* (2008).


**2.2 Surface Forcing**
Surface forcing is derived from the ERA5 reanalysis of *Hersbach, et al.* (2020) by combining three-hourly average estimates
of short and longwave radiative fluxes with six-hourly average estimates of neutral winds at 10m height, 2m air temperature
and humidity, sea level pressure, and daily liquid and solid precipitation.  Surface meteorological variables are converted to
thermodynamic and radiative fluxes within the GFDL Flexible Modelling System coupler, using the Coupled Ocean-
Atmosphere Response Experiment (COARE) bulk formulas (*Fairall et al. 2003*). Incidentally, the complete reanalysis





designation is SODA4.15.2. The '15' in the reanalysis designation refers to our use of ERA5 forcing. The '2' in the
reanalysis designation refers to our use of the COARE bulk formulas.

If ERA5-derived radiative and thermodynamic fluxes were directly applied to the ocean, they would produce systematic
errors in ocean heat and freshwater storage due, we believe, to bias in the meteorological fluxes. We reduce these systematic
errors by carrying out an initial reanalysis during the four-year period 2007-2010. During this initial reanalysis we collect-
information about the misfits in the form of temperature and salinity analysis increments. We then adjust the net surface
heat and freshwater fluxes seasonally and geographically to bring them into closer alignment with ocean heat and freshwater
storage. Previous work by Carton et al. (2018b) shows that this iterative procedure greatly reduces the mean and seasonal
observation-model misfit.
**2.3 Constraining Data**
The main source of subsurface data is the World Ocean Database 2023 of historical hydrographic profiles (*Mishonov et al,*
*2024*) with updates through year 2024. This data set consists of more than 14.4 million profiles during 1980-2024 (**Fig. 5**).
The observation coverage for individual basins is shown in Supplementary Materials **Fig. S7**. In addition to the quality
control procedures carried out by the World Ocean Database we apply ten quality control checks, including checks for
vertical stability, deviation from monthly climatology, extreme misfits to the model forecasts, and, where possible, buddy-
checks. Observations are rejected based on each of the quality control checks but the most important is the comparison to
climatology. Together these additional quality control checks eliminate approximately 10% of the observations.

In recent years more than 13,000 observations per month come from Argo drifting profilers and another ~16,000 per month
come from ocean gliders. Collated L3 remotely sensed nighttime infrared SST observations are obtained from the NOAA
Center for Satellite Applications and Research (*Jonasson et al., 2022a,b*) and are used to update SST. This SST data set is
supplemented by additional calibrating in situ observations obtained from the International Comprehensive Ocean–
Atmosphere Data Set (ICOADS) release 3.0 SST database (*Freeman et al., 2016*). Sea level observations are not
assimilated.



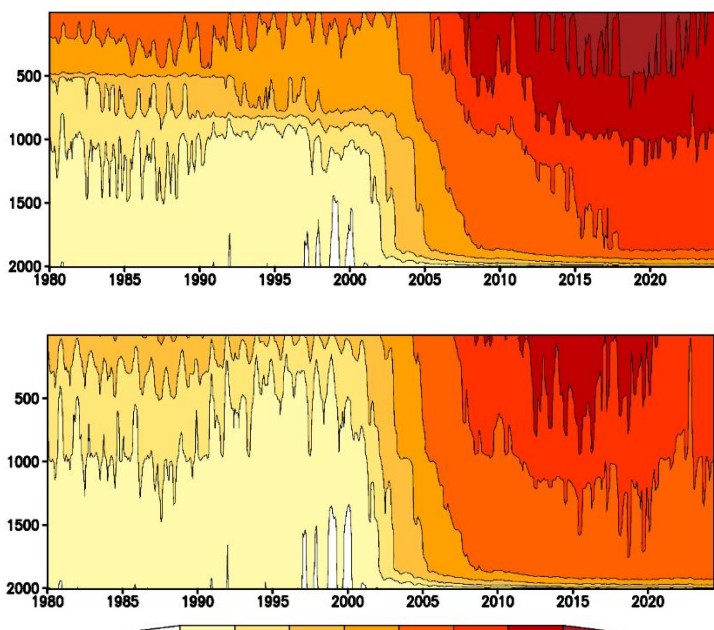

**Figure 5** (upper) temperature and (lower) salinity profiles after quality control is applied, expressed as numbers of observed 1°x1°x10dy cells per year.  Note the dramatic increase in observational coverage following the deployment of the Argo system in the early 2000s.


**2.4 Data assimilation procedure**
SODA uses a simple linear deterministic sequential filter in which the ocean state $\omega^a$ is estimated based on a forecast $\omega^f$
which is adjusted based on the gridded differences between the observations $\omega^o$ and the forecast after bilinear interpolation
to the observation variable locations $\mathbf{H}(\omega^f)$:
$$\omega^a = \omega^f + \mathbf{K}[\omega^o - \mathbf{H}(\omega^f)]$$
(1)


Here the gain matrix $\mathbf{K} = \mathbf{P}^f \mathbf{H}^T \left(\mathbf{H}\mathbf{P}^f \mathbf{H}^T + \mathbf{R}^o \right)^{-1}$ determines the impact of the observations and depends on both the specified
observation error covariance $\mathbf{R}^o \equiv \left\langle \boldsymbol{\varepsilon}^o \boldsymbol{\varepsilon}^{oT} \right\rangle$, which is assumed to be 40% of the signal variance, and a Gaussian model of the
forecast error covariance





$$\mathbf{P^f} \equiv \left\langle \boldsymbol{\varepsilon}^f \boldsymbol{\varepsilon}^{fT} \right\rangle \sim e^{-(\Delta D)^2 / R_D^2 - (\Delta x)^2 / R_x^2 - (\Delta y)^2 / R_y^2 - (\Delta z)^2 / R_z^2 - (\Delta t)^2 / R_t^2}$$
(2)


where the scales ($\Delta D, \Delta x, \Delta y, \Delta z, \Delta t$) are specified following *Carton et al. (2018a)*. Here $\Delta D$ is the change in dynamic
topography whose impact is to allow for extended error covariance along geostrophic streamlines (in other words, allowing
for flow-dependent forecast error). Equation (1) is solved with a 5° localization of the error covariance in both latitude and
longitude.

A direct implementation of (1) would introduce shocks and spurious waves. To avoid this, we use the incremental analysis
update procedure of *Bloom et al. (1996)* with an update cycle of 10 days (chosen to be consistent with the available data and
the time-scale of ocean variability). The formula for **K** is the consequence of minimizing the expected variance of the
analysis error subject to simplifying assumptions, including the assumptions that the model forecast, observation, and
analysis errors are unbiased. Since the observation errors are assumed to be unbiased and uncorrelated $\mathbf{R}^o$ is a diagonal
matrix. The analysis increments, $\mathbf{K}[\omega^o - \mathbf{H}(\omega^f)]$, which are the gridded corrections to the forecast at each assimilation
cycle, are saved and are used to evaluate the degree of misfit between the forecasts and the observations -- negative values
of the time mean temperature analysis increments imply that the model forecasts are biased warm while positive time mean
temperature analysis increments imply that the model forecasts are biased cold. As shown in **Fig. 6** the bias in the upper
ocean is small except within ±5° latitude. Within that latitude band the prevailing winds over the Atlantic and Pacific are
weaker than the model expects, causing the zonal tilt of the thermocline to be too weak both basins. In the Indian Ocean the
zonal tilt is a little bit too strong.

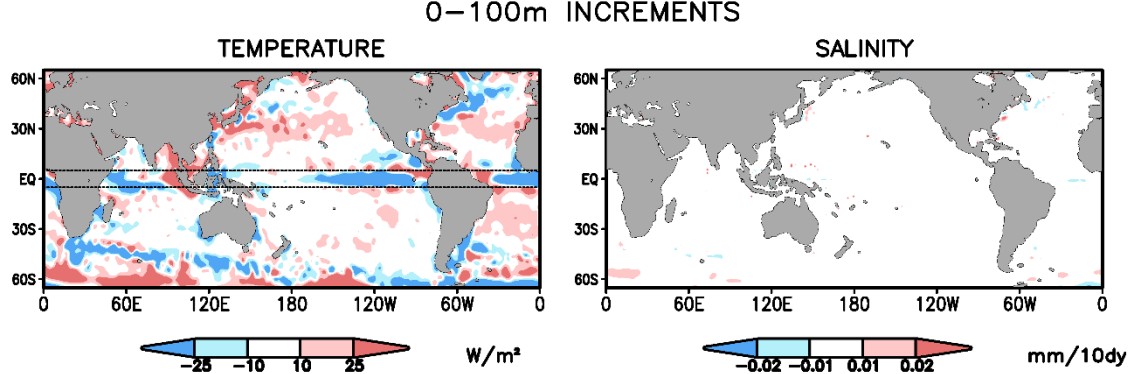





**Fig. 6** Time mean (left) temperature and (right) salinity increments integrated vertically 0-100m and expressed in units of Wm$^{-2}$ and mm/10dy. The 5°S-5°N band of latitudes where wind stress effects dominate are marked with dashed lines.


Additional information about the temperature and salinity analysis increments in the 0-300m and 300m-1000m layers is
shown in Supplementary Materials **Figs. S9-S11**.

At high latitude the Global Ice-Ocean Modeling and Assimilation System (GIOMAS) sea ice thickness estimates of *Zhang*
*and Rothrock* (2003) are used to constrain sea ice thickness. *Schweiger et al.* (2011) optimistically estimate the error in these
thickness estimates to be less than 10 cm in the Arctic. In SODA the net surface heat flux is modified so that ice which is too
thin receives less heat and ice that is too thick receives more heat with a relaxation time-scale of 25 days. Thus, the sea ice
model is allowed to determine the distribution of sea ice among the five snow and ice categories.

SODA4 has produced in four simultaneous streams, each spanning a little more than one decade, similar to the way the
MERRA-2 atmospheric reanalysis was constructed (Gelero et al, 2017). Each stream begins from initial conditions provided
by SODA3.15.2 (**Fig. 7**). Stream1 began on 01 January 1980 and continued to 30 December 1992. Stream 2 began on 31
December 1989 and continued to 02 January 2003. Stream 3 began on 03 January 2000 and continued through 31 December
2015. Finally, Stream 4 began 31 December 2009 and has continued through 31 June 2024. The continuous 45-year record
has been constructed by using the earlier stream during the several years of overlap as indicated by the blue bars.

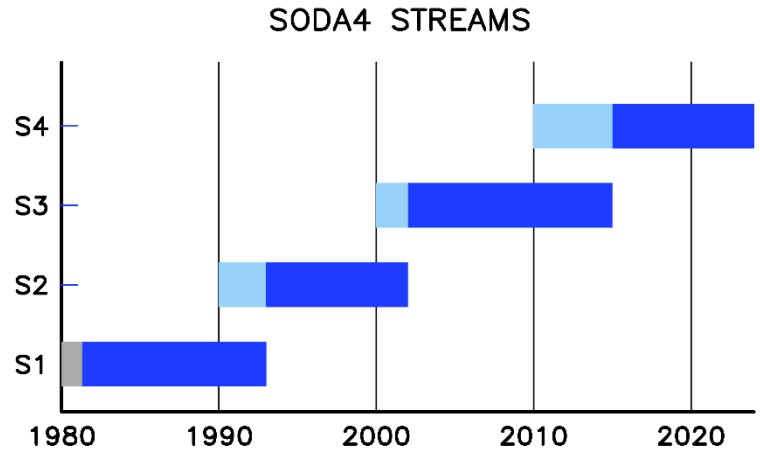





**Figure 7** SODA4 is carried out in four streams beginning 01January, 1980, 01 January 1990, 01 January 2000, and 01 January 2010. Each stream uses initial conditions provided by SODA3.15.2 and extends a few years past the start of the following stream (light blue). The first three years of stream1 which shaded grey indicating that the original files (but not the regridded files) are missing from the archive. The continuous 45-year record concatenates the grey and dark blue sections.


**3. Output Files**

SODA4 output files are accumulated in three directories labelled: ORIGINAL, REGRIDDED, and SODA. ORIGINAL is
divided into three subdirectories: ice, ocean, and transport while REGRIDDED is divided into two subdirectories: ice and
ocean. Each subdirectory contains individual NetCDF4 classic files, written following Climate and Forecast Metadata
Conventions (CF1.4). The file names include the time interval over which the data are averaged, such as 5dy, 10dy, or
monthly, the centering date for that file, and whether it is on the original grid or has been regridded onto a uniform 0.1°x0.1°
lon-lat horizontal grid using bilinear interpolation. Regridded state variables are limited to the upper ocean (0-1017m) for
the years prior to 2005 and to the depth range (0-2006m) beginning in 2005 to reduce file sizes. The file sizes are given in
**Table 1**. The directory SODA contains the temperature and salinity analysis increments at 10dy intervals.

**Table 1** Individual 5dy or 10dy file sizes. The files are written in F32z compressed
NetCDF4 classic format and includes grid information. The total size is approximately
46Tb.

| Grid | Ocean | Ice | Transport |
|---|---|---|---|
| original | 5.3G | 145M | 2.3G |
| regridded | 1.5-1.7G | 32M | |


The contents of the original and regridded ocean and sea ice files as well as the original grid mass transport files are listed in
Supplementary Materials **Tables S2-S5** using standard names and attributes following the CF metadata conventions. A
number of original grid 5dy ocean and transport files, and one 5dy sea ice file are missing from the archive (listed in
Supplementary Materials **Table S6**).. No attempt has been made to fill in the missing files.



### 3.1 Problems

As pointed out above  reanalysis was carried out simultaneously in four streams, one per decade, similar to the way the MERRA-2 atmospheric reanalysis was constructed (*Gelero et al, 2017*).  Carrying out the reanalysis in several streams speed production but has left us with a record that is not quite continuous.  We partially address this issue by extending each stream for several years as illustrated in **Fig. 7**.  The appearance of flow instabilities required us to try several different advective schemes during the integration.  The upwind advection scheme in particular had a noticeable  impact on temperature and salinity increments (Supplementary Materials **Figs. S8-S10**).  Our investigation suggests that the impact of these changes on state variables in the upper ocean is small.

### 4. Results

This section has two parts.  The first part summarizes the results of our comparison of SODA4 temperature and salinity to the monthly EN4.2.2 statistical objective analysis of *Good et al. (2013)*.  The second part of this section presents comparisons of volume transports through key sections in comparison to independent mooring-based observational estimates.

### 4.1 Temperature and Salinity

We begin by comparing the SODA4 subsurface temperature and salinity fields to the EN4.2.2 monthly statistical objective analysis that uses a first guess based on a combination of monthly climatology and persistence.  Since EN4.2.2 lacks input from a numerical forecast model driven by surface meteorology it  lacks the effects of numerical forecast bias but is highly dependent on the observational coverage.  One consequence is that EN4.2.2 has less variability in the Southern Hemisphere than the Northern Hemisphere, and less in earlier decades than later decades.  SODA4 has more geographically and temporally uniform statistics (**Fig. 8 lower,** Supplementary Materials **Fig. S12**).  Near the equator the variability of 0-300m temperature is similar (Supplementary Materials **Figs. S13, S14**).



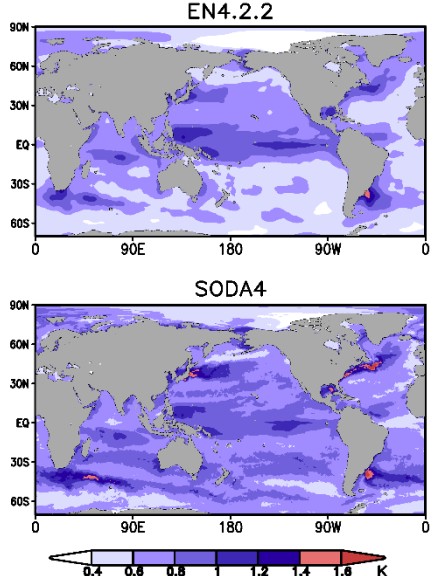

**Fig. 8** Monthly standard deviation of 0-300m
average temperature anomaly from its monthly
climatology (1980-2024). (upper) EN4.2.2, (lower)
SODA4.


Averaged over the whole Pacific basin SODA4 and EN4.2.2 temperatures in the 0-300m and 300-100m layers show a close
correspondence both warming at a rate of 0.12K/10yr (**Fig. 9**) with much of this warming concentrated in the western half of
the basin (**Fig. 10**). Similar correspondence is apparent in the other ocean basins at both depth intervals.



**Fig. 9** Annual and vertically averaged (black) EN4.2.2 and (blue) SODA4 temperature by basin. (left) 0-300m,
(right) 300-1000m.






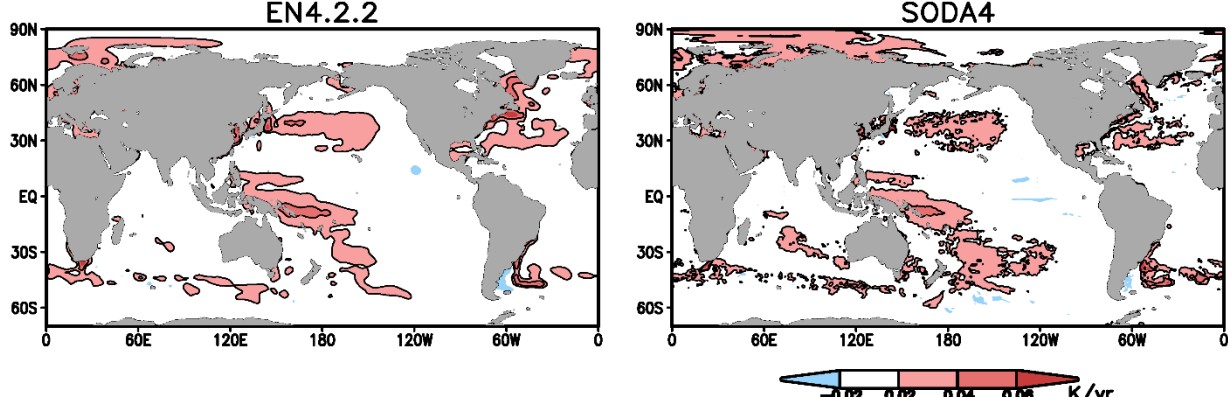

**Fig. 10** Trend in 0-300m temperature during 1981-2023 computed using unweighted least squares. (left) EN4.2.2, (right) SODA4.


EN4.2.2 and SODA4 basin-average salinity in the 0-300m layer shows a similarly close correspondence (**Fig. 11**) . In
contrast, basin-average salinity in the deeper 300-1000m layer shows nonphysical ~0.04 PSU differences. These may be the
result of limited representative salinity observational coverage (Supplementary Materials, **Fig. S7**).





**Fig. 11** Annual and vertically averaged (black) EN4.2.2 and (blue) SODA4 salinity by depth and basin.  (left) 0-300m, (right) 300-1000m.


**4.2 Volume Transports**
We begin by comparing time mean transports through four major passages: Bering, Davis and Fram Straits, and the Barents
Sea Opening at the entrances and exits to the Arctic basin, the Indonesian throughflow between the Pacific and the Indian
Oceans, and Drake Strait cutting across the Antarctic Circumpolar Current (**Table 2**).  SODA4 has a time mean inflow of
$0.86 \times 10^6$ m$^3$/s through Bering Strait, similar to the observed $0.9 \times 10^6$ m$^3$/s.  Transport through Fram Strait has both northward
($6.3 \times 10^6$ m$^3$/s) and southward ($-7.8 \times 10^6$ m$^3$/s) components carrying water both in and out of the central Arctic.  The net flow
through Fram Strait is $1.5 \times 10^6$ m$^3$/s.  The inflow and outflow numbers are both 20% smaller than the one-year inverse model
estimates of *Tsubouchi et al (2018)*, similar to the underestimation reported *by Mayer et al (2023)* for coarser eddy-
permitting reanalyses, but the net outflow is similar to the observed $1.2 \times 10^6$ m$^3$/s..

At the Indonesian throughflow the time mean SODA4 transport from the Pacific into the Indian Ocean of $14.5 \times 10^6$ m$^3$/s is
similar to the observational estimate of $14.7 \times 10^6$ m$^3$/s reported by *Susanto et al. (2016)*.  At Drake Strait (62.4°W) the mean
SODA4 transport of $159 \times 10^6$ m$^3$/s lies between the wide range of published estimates *Koenig et al, 2014; Donohue et al.,*
*2016*).  Time series of annual volume transports through Bering, and Fran Straits and Drake Passage are shown in
Supplementary Materials **Figs. S15-S17**.

**Table 2** Mean (1980-2024) volume transports ($10^6$ m$^3$/s) through four passages, with standard error.

| Passage | SODA4 | Observations |
|---|---|---|
| Bering Strait | 0.9±0.1 | 0.9±0.1[1] |
| Fram Strait (northward) | 6.3±0.2 | 7.4±1.0[2] |
| Fram Strait (southward) | -7.8±0.2 | -8.6±3.6[2] |
| Davis Strait | -1.4±0.2 | -2.1±0.7[2] |
| Barent Sea Opening | 2.6±0.4 | 2.3±3.0[2] |
| Indonesian Throughflow | 14.5±0.4 | 14.7[3] |
| Drake Strait | 159±1.4 | 141 to 173[4] |




[1]Østerhus et al. (2019);[2] Tsubouchi et al. (2018); [3]Susanto et al. (2016); [4]Xu
et al (2020), Koenig et al (2014), Donohue et al. (2016)

Finally, we compare time mean meridional overturning transports across three meridians in the Atlantic: the eastern portion
of the Overturning in the Subpolar North Atlantic Program (OSNAP) section along ~60°N between Greenland and the
United Kingdom (*Lozier et al., 2019*), the Rapid Climate Change-Meridional Overturning Circulation and Heatflux Array
(RAPID) section along 26.5°N (*Moat et al., 2020*), and the South Atlantic Meridional Overturning Circulation (SAMOC)
section along 34.5°S (*Kersalé et al., 2020*) (**Table 3**).  For all three sections the SODA4 meridional overturning transports
agree with the observations to within observational uncertainty. Time series of overturning transport at the RAPID section is
shown in Supplementary Materials **Fig. S18**.

**Table 3** Mean (1980-2024) overturning volume transport ($10^6$ m$^3$/s) across three Atlantic meridians.

| Passage | SODA4 | Observations |
| --- | --- | --- |
| 60°N (OSNAP-East) | 14.2±0.22 | 15.6±0.8[1] |
| 26.5°N (RAPID) | 16.5±0.37 | 17.7±3.7[2] |
| 34.5°S (SAMOC) | 17.2±0.34 | 17.3±5.0[3] |


[1]Lozier et al., (2019); [2]Moat et al., (2020); [3]Kersalé et al., (2020)
**5.  Discussion**
This paper introduces the SODA4 ocean/sea ice reanalysis system which we use to produce the new 45-year long (1980-
2024) SODA4 ocean/sea ice reanalysis.  This ocean/sea ice reanalysis joins two other global eddy-resolving reanalyses:
GLORYS12 (1993-pres, *Lellouche et al, 2021*) produced by the Copernicus Marine Environment Monitoring Service and
BRAN2020 (1994-2016; *Chamberlain et al., 2021*) produced in collaboration between the Australian Department of
Defence, Bureau of Meteorology, and CSIRO.  SODA4 is built using GFDL MOM5/SIS1 numerics with 0.1°x0.1°
horizontal resolution and 75 Z* levels in the vertical. ERA5 meteorological estimates provide sub-daily surface forcing
while recently updated World Ocean Database hydrographic observations, NOAA in situ and satellite SST data sets, and
GIOMAS ice thickness estimates are all used as constraints within the sequential data assimilation reanalysis algorithm.  A
new monthly discharge data set has been developed specifically for this reanalysis, building on previous work by *Dai and
Trenberth (2020)*.  Reanalysis state variables as well as some ancillary variables such as mixed layer depth are available on



both the original grid and remapped onto a uniform 0.1°x0.1° lon-lat horizontal grid.  The size of the total data set is
approximately 46Tb.
A limited number of comparisons are presented here to evaluate system performance.  Temperature and salinity bias,
evaluated by examination of the mean gridded observation-minus-forecast differences, is shown to be low except within a
few degrees of the equator where slightly weak ERA5 trade winds introduce an artificial zonal gradient in forecast
temperature.  The month-by-month accuracy of temperature and salinity is evaluated by comparison to the EN4.2.2
statistical objective analysis in two upper ocean layers: 0-300m and 300-1000m. EN4.2.2 is constructed using an archive of
historical temperature and salinity observations that is similar to the World Ocean Database, but without use of a numerical
forecast model and thus lacks that potential source of forecast bias.  Basin-average comparisons show that temperatures are
quite similar and salinities are also similar in the 0-300m layer but less so in the 300-1000m layer.  The latter improves after
Argo observations become available in the early 2000s except in the Arctic basin which is not sampled by Argo.  The
SODA4-EN4.2.2 comparison also makes clear that the EN4.2.2 analysis is strongly sensitive to the distribution of
observations.  This dependence introduces an artificial trend in variability which is most noticeable in the Southern
Hemisphere.
Time-mean volume transports are evaluated at a limited number of major passages separating ocean basins and also through
several Atlantic basin-spanning transects in comparison to moored observations.  At the Pacific gateway to the Arctic
SODA4 transport through Bering Strait is consistent with observations.  At the Atlantic gateways to the Arctic net transport
through Fram Strait, Davis Strait, and the Barents Sea Opening are also consistent with independent estimates (*Tsubouchi et*
*al.*, 2018).  At the Indonesian Throughflow and at Drake Strait in the Southern Ocean mean transport is also consistent with
the range of reported observational estimates (a wide range in the case of Drake Strait).  Finally, we examine the mean
SODA4 transport along three transects monitoring the strength of the Atlantic meridional overturning circulation.  Again, the
mean values of SODA4 overturning volume transport across the eastern subpolar OSNAP line (60°N), the northern
subtropical RAPID line (26.5°N), and the southern subtropical SAMOC (34.5°S) line all agree to within reported
uncertainties.
Two complications arose while producing this reanalysis.  The first was the result of an effort to speed production by
dividing the SODA4 reanalysis into four simultaneous streams.  To reduce discontinuities each stream was extended for
several years of overlap.  The second complication was the need to change from a third order upwind tracer advection
scheme to one of the monotonicity-preserving schemes of *Daru and Tenaud (2003)* during the integration.
With its resolution improvements the SODA4 reanalysis is now able to represent processes such as those controlling
development and movement of fronts and eddies more accurately than its eddy-permitting predecessors throughout most of



the ocean. One region where even the soda4 resolution is insufficient is the Arctic. Accurate reanalyses of Arctic Ocean
dynamics will need even greater improvements to resolution, perhaps aided by exploiting machine learning analogues to
represent the contributions of the unresolved scales of motion.





**Code and data availability**

The GFDL MOM5/SIS1 model is a widely used code. Detailed documentation regarding GFDL MOM5/SIS1 is available
on GitHub. Likewise, the SODA4 active repository is accessible on github at https://github.com/UMD-AOSC/soda4. We
have uploaded the source codes of both to Zenodo (https://doi.org/10.5281/zenodo.16878109). Scripts to produce the
images are also archived on https://zenodo.org/records/16934040. Since this is a reanalysis project it relies on many input
data sets. Current versions of the hourly and monthly ERA5 data sets are available through the Climate Data Store
(ecmwf.int), the World Ocean Database collection of subsurface observations are available through
https://www.ncei.noaa.gov/products/world-ocean-database, the NOAA Office of Satellite and Product Operations L3
satellite SST observations are available through https://www.ospo.noaa.gov, the UK Met Office EN4.2.2 analysis of
subsurface temperature and salinity profile observations are available through
https://www.metoffice.gov.uk/hadobs/en4/download-en4-2-2.html, the Arctic Great Rivers Observatory discharge
information are available through https://www.arcticrivers.org/data, and the Global Ice-Ocean Modeling and Assimilation
System (GIOMAS) data are available through https://psc.apl.washington.edu/zhang/Global_seaice/data.html. The ocean/sea
ice reanalysis files described in this paper are accessible through https://dsrs.atmos.umd.edu/DATA in four subdirectories:
soda4.15.2_s1, /soda4.15.2_s2, soda4.15.2_s3, and soda4.15.2_s4.

**Author contribution**

JAC provided overall guidance and much of the initial writing. GAC managed the data sets, carried out all integrations. JAC,
GAC, LS, and SGP all contributed to analysis and general improvements to the manuscript.

**Competing interests**

The authors declare that they have no conflict of interest.



**Acknowledgements**
We gratefully acknowledge the producers of the ECMWF ERA5, the UK Met Office EN4.2.2 analysis, the Applied Physics
Laboratory PIOMAS analysis, the contributors to and managers of the World Ocean Database collection of hydrographic
profiles, and the NOAA Office of Satellite and Product Operations satellite SST observations.  Computer resources were
provided by the National Center for Atmospheric Research.
**Financial support**
JAC and GAC were supported by the National Science Foundation (OCE5235332).  LS was supported by NOAA Grants
NA24NESX432C0001 and NA19NES4320002 [Cooperative Institute for Satellite Earth System Studies (CISESS)] at the
University of Maryland/ESSIC.  SGP was partially supported by NOAA Weather Program Office award NA23OAR4590196-
T1-01 and by NASA awards 80NSSC23K0827 and 80NSSC24K1506.

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
