# Peer review of "SODA4: a mesoscale ocean/sea ice reanalysis 1980-2024"

_EGUsphere, 2025_

## Community Comment (CC3)

The manuscript describes the configuration of the new ocean reanalysis SODA4, which in comparison to its predecessor SODA3 features higher horizontal and vertical resolution, and which additionally assimilates sea ice data. The reanalysis is evaluated in terms of temperature and salinity changes, as well as transports at a few key locations.

SODA is a widely used product and updates are useful and descriptions of those are necessary.

The presented description/evaluation is very similar to those presented for previous products.

However, I am missing details of a comparison to the previous product SODA3, which would allow the reader to assess the level of advancement. Only the Gulf Stream paths are intercompared for this purpose.

The goals of this paper are somewhat different than the goals outlined by the reviewer. The goals for this paper are: 1) to document this mesoscale ocean/sea ice reanalysis, and 2) to examine its basic performance. These goals are similar to those of Lellouche et al. (2021, https://doi.org/10.3389/feart.2021.698876) in their description of GLORYS12. The difference is that we also add comparisons to independent observations (e.g. section transports shown in Tables 2 and 3, Fig. S18 and sea level, Fig. S19).

The reviewer asks about our comparison to the widely cited EN4. Our motivation for comparison to EN4 is that EN4 explicitly lacks that part of the bias which comes from use of a dynamical forecast model. We think the reason why climate documents such as the Gulev et al. (2021) IPCC report only rely on products similar to EN4 and ignore the reanalyses is because of their concern about dynamical forecast model bias.

This reviewer asks for additional comparison of SODA4 to the ¼-deg reanalyses such as SODA3 and ORAS5. We do think such a study will be interesting. The results of such a study will be strongly regional. For example, a student, Shaun Eisner, has already produced three such studies for the Arctic alone (e.g. Eisner et al., 2025 https://doi.org/10.5194/egusphere-2025-5737). Similar comparisons could be carried out in many region to look at the impacts of resolution. But here we focus on dynamical forecast model bias.

For temperature and salinity, the manuscript also offers only an intercomparison to the objectively analysed EN4 product instead of an evaluation with the EN4 or other in situ data.

We think the reviewer is asking for a comparison to the EN4 or WOD observation sets. The data assimilation cycling carried out in SODA4 automatically produces comparison statistics in the form of temperature and salinity analysis increments. Presentation of the

analysis increments is included in the discussion of Fig. 6 (lines161-170), and in supplementary materials (see Figs. S8-S10).

Based on only intercomparisons, the evaluation of advancement and quality of the product is difficult.

As discussed above the goal of this paper is not to present a comparison of SODA4 vs SODA3. The paper goals are: 1) to document this mesoscale ocean/sea ice reanalysis, and 2) to examine its basic performance. Our evaluation of performance, similar to what is presented in Lellouche et al., begins with an examination of the analysis increments. We extend this examination to consider how similar SODA4 is to a variety of independent observations, covered both the main text and in the extensive supplement.

EN4 objectively analysed data is flawed in many ways, as it is also acknowledged in the manuscript. Therefore, the purpose of the intercomparison becomes ambiguous. It may either show the advantage of using a dynamical model for an analysis or an evaluation of the SODA4 analysis. I suggest to include a comparison to the EN4 in situ data and the previous SODA3 product as it has been done before. The purpose of the intercomparison to the EN$ analysis could be clearer. More details are below.

We point out that the main text is already 6695 words long with three tables and 11 multi-panel figures while the Supplement adds an additional 2376 words, seven tables, and 19 multi-panel figures. We think additional comparisons belong in separate papers that can focus on, and interpret, regional differences (such as Eisner et al., 2025 https://doi.org/10.5194/egusphere-2025-5737).

Details

l 49-50 I cannot find anything about discharge in this reference. Please clarify or correct. Reference moved to the middle of the sentence to prevent confusion. The sentence now reads: "*SODA4 has been developed to address these limitations by increasing model resolution to 0.1°x0.1°x75L, upgrading the observation sets to include the 18.6 million profiles contained in the World Ocean Database 2023 (Mishonov et al., 2024), and adding improved estimates of continental discharge*".

l 50-51 I suggest to include two sentences about what Fig. 1 shows. Maybe a comparison to observations could be included to provide an idea about the achieved realism. We have one sentence highlighting one of the impacts of enhanced resolution (L50-51). We think that if we try to elaborate, we will be led into a more extended comparison of SODA3 and SODA4, which is not the purpose of this paper.

l 60-62 How about the Southern Ocean https://doi.org/10.1175/JCLI-D-14-00353.1 describes 1/10 as resolving equator ward of 50N/S

Sentence modified to note a similar problem at southern latitudes.  The sentence now reads:

> "For much of the global domain this resolution satisfies the requirements of Hallberg (2013) to be considered 'eddy resolving' but reduces to 'eddy permitting' in the Arctic and the deep Southern Ocean due to the decrease in the eddy length-scale".

l 86-89 Why would this be a good idea ? Is there a reason to believe that the evaporated water is brought back by the rivers into the same basin? Why evaporation and not E-P? How does this bring the flux in alignment with the salinity trends The Atlantic is believed to import freshwater, It seems you put a constrain on this which would rather be something the model should deliver you., but I don't understand what has been done.
l 88-89 I am missing more details how it was carried out, S1 states the same details about the balancing as provided here.

We have modified the main text (L. 89-92) to clarify that our continental discharge adjustments to account for ungauged discharge follow those described in Dai and Trenberth (2002), who in turn refer to the work by Fekete et al. (2000) using the ECMWF reanalysis to calculate gauged/ungauged ratios. Like Dai and Trenberth, we find that the Fekete et al ratios give reasonable global continental discharge estimates.  The supplemental materials text S1 has also been modified to be consistent with this description.  The paragraph in S1 now reads:

> To address this problem we first attempt to find alternate sources of monthly discharge for major river systems.  Fig. S4 shows the Dai and Trenberth record of monthly discharge for the Ob and Yenisei rivers in comparison to the SODA4 discharge updated based on data from arcticgreatrivers.org (Shiklomanov et al., 2020).  We also add monthly discharge from ice sheets such as Greenland which are entirely missing from Dai and Trenberth (Fig. S5).  Many gaps remained after this initial exercise.  The remaining gaps we fill with climatological monthly values, inflating the implied discharge rates to account for the ~30-40% of the discharge that Dai and Trenberth (2002) suggest is ungauged.  The final SODA4 monthly time series of global discharge is shown in Fig. S6.  Its time mean, 1.14x106 m3/s, falls at the lower end of the span of recently published estimates: 1.1-1.4x106 m3/s (Syed et al., 2010; GRDC, 2014; Wilkinson et al., 2014; Müller-Schmied, et al. 2014).

l 146 Aren't the Rs the scales and the deltas the distances ?
Thank you.  Fixed

l 170-171 How is the heat flux modified, via a relaxation term depending on the sea ice thickness difference to the data?
The relaxation term is described in (L177)

> "In SODA the net surface heat flux is modified so that ice which is too thin receives less heat and ice that is too thick receives more heat with a relaxation time-scale of 25 days."

Section 3 I think details of the output files would be better presented on the web page that provides the data, best with a doi.

This was also suggested by another reviewer.  The text has been moved to Supplementary Materials, text S2.

Section 4.1 As EN4 may have larger errors relative to than SODA4, it is not clear what this comparison evaluates. Particularly in sparsely observed locations EN4 often falls back to climatology.
As discussed in the responses to reviewer's comments above, a major motivation for our comparison to EN4 is quantification of bias and specifically identification of the role of the dynamical forecast model in introducing bias.  We attempt to clarify this in our discussion of the comparison to EN4 in 5. Discussion, L289-300.

Is the comparison an illustration of what additional information SODA4 offers or and evaluation? It would be good to give the reader some guideline.

We attempt to summarize the distinction between SODA4 and the widely cited EN4 and to describe what additional information has been gained from the comparison to SODA4 in L. 293-297:
>    EN4.2.2 is constructed using an archive of historical temperature and salinity observations that is similar to the World Ocean Database, but without use of a numerical forecast model and thus lacks that potential source of forecast bias.  Basin-average comparisons show that temperatures are quite similar and salinities are also similar in the 0-300m layer but less so in the 300-1000m layer.  The latter improves after Argo observations become available in the early 2000s except in the Arctic basin which is not sampled by Argo.

Figures Why sometimes temperature is K and sometimes in ˚C? Changes are sometimes in W^2/m and sometimes in ˚C/yr
The units are chosen to reflect how we think the users will use different results.  For example, Fig. 6 shows time mean vertically integrated temperature and salinity increments. These are expressed in units of $W/m^2$ and mm/dy so that users can estimate the implied bias in net surface heat flux ($W/m^2$) and freshwater flux (mm/dy). Figs. 8-10 shows temperature deviation, temperature, and temperature trend  in K and K/yr because that seems the units most commonly used for such figures.

l 250 It says Drake Strait in Fig. S17
Changed to 'Drake Passage' throughout the text and supplementary material.

l 262-265 Given that a large part of the AMOC variability is related to Ekman transport and resulting in RAPID and SODA4 from ERA5 the agreement between RAPID and SODA4 is surprisingly low. Is the Florida Strait transport or the upper midocean transport responsible for this?

We think the reviewer is referring to supplemental materials figure S18 which compares time series of AMOC overturning transport.  In contrast the main text only presents the time

mean).  We have added a brief note to the figure S18 legend indicating the SODA4 overturning transport may be impacted by the change in advection scheme prior to 2015.

Here is the revised Legend:
**Fig. S18** Atlantic meridional overturning circulation across 26°N ($10^6$ m$^3$/s).  (blue) SODA4, (black) RAPID time series. The time mean SODA4 overturning transport is: 16.5±0.4 x$10^6$ m$^3$/s. The overturning transport may be impacted by the change in advection scheme prior to 2015 (see Fig. S11b).

Table 3 How is the overturning defined. Is it at a fixed depth (which?) or the maximum? And is is in density or z-coordinates?

A clause has been added to L253-5 giving our definition of the overturning transport
Finally, we compare time mean meridional overturning transports (defined as a maximum of the zonal integrated  stream function) across three meridians in the Atlantic.

l 282 293 I think for an evaluation of SODA4 a comparison to the actual in situ temperature is required. With the last sentences you seem to evaluate EN4 with SODA4. It's odd to change the reference based on occasion.

As discussed at the bottom of the first page of this response to reviewer 3 the temperature and salinity analysis increments are the misfits with respect to the  in situ temperature and salinity  and they are discussed in the text surrounding Fig. 6 as well as supplemental materials Figs. S8-S10.

L 313 SODA4
fixed.